# HR Managers’ Gender and Rationality Culture: Interaction Effects on Female Employees’ Workplace Outcomes

**DOI:** 10.3390/bs15081088

**Published:** 2025-08-12

**Authors:** Maftunakhon Utkir kizi Tojimatova, Soo Young Shin

**Affiliations:** School of Business, Yeungnam University, Gyeongsan 38541, Republic of Korea; ladym1@yu.ac.kr

**Keywords:** human resource managers’ gender, rationality culture, female employees, interpersonal affinity, organizational loyalty, job involvement

## Abstract

This study investigates how the gender of human resource (HR) managers and the presence of rationality culture (RC) in organizations jointly influence women employees’ workplace outcomes, including interpersonal affinity, job involvement, and organizational loyalty. Drawing on feminist organizational theory and social identity theory, the study examines whether women HR managers are associated with more positive outcomes for women employees and whether the dominance of RC moderates these effects. RC, rooted in bureaucratic logic and objectivity, may diminish the perceived value of relational and inclusive HR practices—especially in highly formalized work environments. The study employs a survey-based quantitative method using data from the Korean Women Managers Panel, which includes responses from over 346 women working in medium- and large-sized organizations in South Korea. Directional hypotheses are tested, proposing that women HR managers positively influence employee outcomes, but this effect may be weakened in organizations where RC is strongly embedded. The findings contribute to organizational behavior and the gender studies literature by clarifying how HR managers’ gender operates under varying cultural norms and revealing the conditional nature of its effectiveness. The study offers both theoretical and practical insights for organizations aiming to foster inclusive environments, with implications for HR strategy, organizational development, and gender

## 1. Introduction

In recent decades, gender diversity in management has emerged as a critical focus in both academic discourse and organizational practice, particularly within the field of human resource management (HRM). As the number of women occupying managerial roles continues to rise, questions about the legitimacy and effectiveness of female managers persist ([5]; [14]). Despite these numerical gains, traits traditionally associated with women—such as empathy, collaboration, and relational communication—are often undervalued in organizations dominated by bureaucratic and rationalist cultures ([3]). This paradox highlights the need for deeper investigation into how the gender of human resource (HR) managers influences female employees’ perceptions of fairness, trust, and psychological safety. While prior research has examined gender representation in broader leadership and supervisory roles (e.g., [5]), the intersection of HR leadership and organizational rationality culture remains significantly underexplored.

This study builds on feminist organizational theory as the primary framework and integrates perspectives from social identity theory and role congruity theory to construct a multi-level understanding of gendered leadership in structured organizational settings. Feminist organizational theory critiques the structural and cultural biases that systematically devalue femininity and emotional labor, particularly within formalized institutions ([1]). In this context, female HR managers may be symbolically present but remain structurally excluded from influence and decision-making, especially in environments shaped by rationality culture (RC).

Social identity theory ([19]) provides a micro-level lens to understand how shared gender identity between female HR managers and female employees can foster interpersonal affinity, emotional trust, and psychological safety. Role congruity theory ([6]) further explains how expectations about gendered behavior may facilitate or hinder perceptions of leadership effectiveness. Together, these theories suggest that female HR managers may positively influence emotional and organizational outcomes among female employees—but only under certain organizational conditions.

One such condition is the degree to which organizations are embedded in rationality culture (RC)—an orientation characterized by formalism, control, strategic detachment, and depersonalized decision-making ([21]; [7]). RC is rooted in classical management theory and often rewards traits such as hierarchy, logic, and objectivity, which have historically been associated with masculinity. Recent studies have begun to empirically explore how rationality-oriented cultures constrain the strategic influence of HR professionals. For instance, prior studies have found that HR managers in bureaucratic Korean organizations faced limited autonomy in decision-making due to rigid procedural frameworks. Similarly, research show that efforts to implement inclusive HR initiatives are often deprioritized in environments that privilege formalism and quantifiable outputs. These empirical insights support the idea that rationality culture not only shapes organizational routines but also restricts the relational and emotional capacities typically associated with female HR leadership. This growing line of research underscores the need to examine how such structural conditions interact with gendered leadership in HRM. Feminist theorists argue that in such environments, competencies linked to emotional intelligence, relational support, and inclusivity—often attributed to female leaders—are systemically undervalued ([1]; [7]).

Empirical studies suggest that in highly formalized organizations such as Korean public institutions or large conglomerates (e.g., chaebols), HR practices are governed by strict procedural norms and hierarchical control systems ([15]). In these contexts, women HR managers frequently encounter resistance when attempting to implement relational or inclusive HR strategies ([10]; [16]). For instance, initiatives such as mentorship programs or empathy-based conflict resolution may receive limited institutional support in environments that prioritize efficiency and quantifiable outcomes over employee well-being.

This practical tension illustrates the critical need to understand how the structural logic of RC interacts with the relational strengths commonly associated with female HR leadership. While the previous literature acknowledges the symbolic value of increasing female representation in HRM, our study introduces a novel theoretical insight: gender-congruent HR leadership does not operate uniformly across organizational contexts. Specifically, we demonstrate that the effectiveness of female HR managers in fostering job involvement, interpersonal affinity, and organizational loyalty among women employees is conditional upon the cultural orientation of the organization. This challenges the common assumption that representation alone ensures inclusion and underscores the importance of aligning institutional culture with diversity initiatives.

By addressing this gap, the present study empirically examines how the gender of HR managers and rationality culture jointly shape female employees’ workplace experiences. It asks whether gender congruence in HR leadership improves trust, engagement, and commitment—and whether RC moderates these effects.

Theoretically, this study contributes to the literature by integrating feminist organizational theory with social identity and role congruity theories, offering a multi-level explanation of gendered HR leadership in structured settings. Unlike prior studies that treat gender representation as universally beneficial, we adopt a culturally contingent perspective, showing that structural logics can either amplify or suppress the benefits of shared gender identity in leadership.

Practically, the findings offer guidance to HR strategists and policymakers: appointing female HR managers alone may be insufficient unless organizational cultures also evolve to value relational competencies. This has direct implications for leadership development, diversity strategies, and institutional design.

The remainder of the paper is organized as follows. The literature review outlines the theoretical foundations of rationality culture and feminist critiques of gender in HR leadership. It then explores the interaction between gender congruence and organizational culture in shaping employee outcomes. Finally, the conceptual model and hypotheses are presented to guide the empirical analysis.

### 1.1. Literature Review and Hypotheses Development

#### 1.1.1. RC

RC, as theorized by [21] ([21]), is rooted in the foundational model of bureaucracy that prioritizes formal rules, procedural consistency, and logical reasoning as mechanisms for organizational control and efficiency. Within this framework, organizations are structured to eliminate ambiguity and subjectivity in decision-making. Although this model contributes to standardized procedures and accountability, it is also embedded with certain cultural values, especially those that align with traditionally masculine traits such as objectivity, autonomy, and authority. These characteristics, although useful in many contexts, inherently marginalize alternative approaches that emphasize interpersonal responsiveness or emotional connection.

Feminist organizational scholars have critiqued this model by highlighting how RC is not truly neutral but systematically favors behaviors and attributes socially regarded as masculine ([13]; [8]). Emotional detachment, competitiveness, and command control hierarchies are upheld as professional ideals, whereas collaboration, empathy, and contextual sensitivity—traits often attributed to women—are regarded as secondary or even antithetical to authority and organizational influence. This results in a structural devaluation of contributions associated with feminine identity, especially in environments where bureaucratic control is deeply entrenched.

This conflict is especially pronounced in HRM. HR roles typically require negotiation, support, and emotional labor, functions that rationalist paradigms tend to overlook or de-emphasize. When HR professionals, particularly women, rely on relational or supportive approaches, they may be perceived as lacking the assertiveness or objectivity deemed essential in rationalist settings. This perceived misalignment can create barriers to perceived legitimacy and authority for women in HR roles, especially in contexts that elevate formalism over human-centric skills.

RC may moderate the effectiveness of HR managers’ gender on employee perceptions. In highly rationalistic settings, the supportive strengths associated with female HR managers may be undervalued, leading to diminished employee perceptions of their warmth, strategic credibility, and trustworthiness. This perceived misalignment can create barriers to perceived legitimacy and authority for women, particularly in HRM roles where strategic authority is already limited. For instance, [12] ([12]) found that female HR professionals in formalized organizations often experience skepticism regarding their decision-making capacity due to expectations of emotionality or supportiveness

While rationality culture (RC) has been conceptually linked to bureaucratic logic and classical management theory, its direct application to the HRM domain remains underexplored. To address this gap, the revised manuscript incorporates the prior literature that indirectly examines the rationalization of HR processes, such as that by [10] ([10]), who discusses the gendered consequences of rationalism in HR, and that by [16] ([16]), who investigates how the feminized image of HR impacts decision-making authority within structured organizations.

However, few empirical studies explicitly examine how RC moderates the influence of HR managers’ gender on employee outcomes. This study addresses that void by operationalizing RC as a contextual variable and testing its interaction with gender representation in HRM roles. This empirical application expands the RC construct beyond general organizational structure and into the realm of human resource leadership, thereby extending both theoretical and empirical boundaries of prior work. Additionally, it contributes to feminist organizational scholarship by embedding gender critiques within the functionally essential, yet structurally marginalized, field of HRM.

#### 1.1.2. Feminist Organizational Theory

Feminist organizational theory provides a powerful lens for understanding how formal structures and cultural norms reinforce gender hierarchies in organizations. As [1]’s ([1]) theory of gendered organizations posits, organizational roles, job descriptions, and evaluation metrics are not gender-neutral. However, they are embedded with values and expectations that advantage men and masculinized behaviors. Gendered assumptions about who is a “natural” authority figure persist in many workplaces, resulting in a bias toward male-dominated leadership even when women are present in significant numbers. For example, [17] ([17]) documents the persistence of male-dominated hierarchies even in feminized professions, while [9] ([9]) find that women leaders often must “prove” their legitimacy in ways not expected of men, particularly in formalized work settings.

In the context of HRM, these dynamics are further complicated by the profession’s feminized identity. According to [22] ([22]) and [11] ([11]), HR roles often fall within the domain of emotional labor and caretaking—qualities socially assigned to women and devalued in comparison to task-oriented, results-driven functions. Thus, although women may numerically dominate HR departments, their authority is often perceived as limited to “soft” domains, distancing them from high-stakes decision-making.

The phenomenon of symbolic representation further challenges the real power of women in HR. [7] ([7]) described how the presence of women in managerial roles may be used by organizations to signal gender inclusivity without providing actual authority or autonomy. In such cases, female HR managers may be expected to champion diversity and empathy while being systematically excluded from core strategic conversations. This disconnect can create disillusionment among female employees, who may initially perceive female HR managers as allies but later recognize the institutional constraints limiting their influence.

This perceived lack of real authority may directly affect employee perceptions. When female HR managers are perceived as symbolic or powerless, female employees may question their capacity to advocate effectively on their behalf. This can lower employees’ feelings of interpersonal affinity, reduce motivation or job involvement, and weaken organizational loyalty. Feminist critiques thus reveal how structural barriers undermine not only women’s formal authority but also the trust and engagement they can generate among other women in the workplace.

#### 1.1.3. Social Identity Theory

Social identity theory ([19]) posits that individuals categorize themselves and others into social groups, deriving part of their self-concept from group membership. In workplace settings, shared identity between employees and managers—such as gender similarity—can foster interpersonal trust, mutual support, and psychological safety. This identity congruence enhances perceived empathy and reduces social distance, particularly beneficial for marginalized groups such as women in male-dominated environments

#### 1.1.4. Integration of Feminist Organizational Theory and Social Identity Theory

The integration of feminist organizational theory and social identity theory is essential to fully account for the multi-dimensional nature of employee outcomes in gendered organizational settings. Feminist organizational theory provides a macro-level critique of structural and cultural biases that systematically marginalize women’s contributions, particularly in contexts where rationality culture dominates. It explains why female HR managers may be symbolically included but structurally excluded from real authority, thus limiting their influence despite representation.

In contrast, social identity theory operates at the micro-level, explaining how shared gender identity between female HR managers and female employees can foster emotional resonance, psychological safety, and trust. This theory accounts for how interpersonal affinity and motivation arise from perceived identity alignment, particularly in environments where women are underrepresented or marginalized.

Together, these theories offer a complementary framework: feminist theory explains the institutional constraints, while social identity theory explains the interpersonal mechanisms. Their integration allows for a richer, more holistic understanding of how HR managers’ gender impacts female employees’ perceptions—and under what organizational conditions these impacts are amplified or suppressed. This theoretical synthesis strengthens the explanatory power of our model and justifies the development of interaction-based hypotheses that consider both identity-based alignment and structural-cultural constraints.

#### 1.1.5. HR Manager Gender and Organizational Outcomes

Building on feminist and organizational theories, the gender of an HR manager may play a critical role in shaping key organizational outcomes. One important domain is organizational loyalty, which reflects the extent to which employees are emotionally committed to their organization and willing to contribute to its long-term success. Research suggests that through their perceived emphasis on fairness and support, female HR managers may help foster loyalty by creating inclusive and values-aligned workplace cultures ([14]; [5]). These qualities are strongly linked to organizational commitment and reduced turnover intentions ([4]).

Second, job involvement is influenced by workplace dynamics that promote psychological engagement and perceived value in one’s work. Female HR managers may be perceived as more encouraging of collaboration and recognition of individual contributions, which enhances employees’ intrinsic motivation and their sense of purpose in an organization ([18]). Thus, HR managers’ gender may be associated with greater job involvement, particularly in settings that support inclusive practices.

Third, interpersonal affinity, defined as the perception of warmth, empathy, and approachability, is often enhanced when employees share identity-relevant characteristics with their managers. According to social identity theory, shared gender between female employees and female HR managers can strengthen interpersonal resonance and emotional connection ([19]). While early studies, such as that of [7] ([7]), noted that the presence of women in HR can serve symbolic functions, more recent research confirms that this pattern persists. For instance, [12] ([12]) found that organizations frequently promote women to HR positions to signal gender inclusivity and responsiveness, even when decision-making power remains centralized among male executives. This practice, often referred to as “optical diversity,” risks reinforcing existing hierarchies while giving the appearance of progress.

These insights highlight how HR managers’ gender can shape multiple organizational outcomes, including loyalty, job involvement, and interpersonal affinity. Therefore, the following hypotheses are proposed:

**H1.** 
*The gender of an HR manager influences employees’ organizational loyalty; specifically, female HR managers are associated with higher outcomes among female workers than male HR managers.*


**H2.** 
*The gender of an HR manager influences job involvement; specifically, female HR managers are associated with higher outcomes among female workers than male HR managers.*


**H3.** 
*The gender of an HR manager influences interpersonal affinity; specifically, female HR managers are associated with higher outcomes among female workers than male HR managers.*


#### 1.1.6. Interactive Effect of HR Managers’ Gender and RC

Prior research suggests that the positive influence of female HR managers on employee perceptions may depend not only on shared gender identity but also on the organizational context in which they operate. Although gender congruence between female employees and female HR managers can strengthen perceptions of trust, fairness, and belonging ([5]; [19]), the broader cultural environment may either amplify or constrain these effects.

As previously described, RC is grounded in formalism, procedural logic, and strategic detachment. These characteristics may suppress the relational or inclusive contributions typically associated with women in HR roles. In such settings, even when female HR managers foster interpersonal affinity, their positive influence on formal outcomes such as organizational loyalty or job involvement may be dampened by institutional norms that prioritize objectivity and procedural efficiency ([1]; [7]). Thus, the effectiveness of an HR manager’s gender in shaping employee attitudes is not only a matter of identity alignment but also conditioned by how an organization values or marginalizes relational and emotional contributions.

Drawing from these theoretical perspectives and reflecting on the structure of the proposed research model, the following directional hypotheses are proposed:

**H4.** 
*RC moderates the relationship between an HR manager’s gender and female employees’ organizational loyalty such that the positive effect of a female HR manager is weaker in highly rationalistic cultures.*


**H5.** 
*RC moderates the relationship between an HR manager’s gender and female employees’ job involvement such that the positive effect of a female HR manager is weaker in highly rationalistic cultures.*


**H6.** 
*RC moderates the relationship between an HR manager’s gender and female employees’ perceived interpersonal affinity such that the positive effect of a female HR manager is weaker in highly rationalistic cultures.*


These hypotheses capture both the direct impact of an HR manager’s gender on employees’ perceptions and the conditional role of organizational culture, offering a compact but theory-aligned framework for empirical testing.

## 2. Method

### 2.1. Sample and the Procedure

This study used the Korean Women Managers Panel Data, surveyed and published by the Korean Women’s Policy Institute. Comprising 769 companies, the data have been collected biennially since 2007 to understand the employment status, career development, wages, and welfare benefits of female workers in Korean companies. These data target companies that employ at least one female manager and have 100 or more employees. In this study, we used organization-level data from the 2018 wave of the Korean Women Managers Panel, and after excluding cases with missing values, the final analysis was conducted on 346 organizations. The descriptive statistics of the sample are as follows:-Firm size: a total of 17 organizations had 100–299 employees, 157 had 300–999 employees, 91 had 1000–1999 employees, and 35 had 2000 or more employees.-Organization type: a total of 301 organizations were in the private sector, and 45 were public organizations.-Gender of HR managers: in the sample, 201 HR managers were male, and 145 were female.

Hierarchical regression analyses were conducted to test the hypothesized relationships, controlling for organization type and size. All statistical analyses were performed using IBM SPSS Statistics version 29. The rationale for using hierarchical regression was to examine both main effects and interaction effects (moderation) while accounting for covariates.

### 2.2. Measurements

The participants assessed employees’ outcomes (loyalty, involvement, and affinity) on a five-point Likert scale ranging from 1 to 5. For instance, regarding loyalty, the following options were provided: “Male employees demonstrate significantly higher levels of loyalty than female employees”, 1; “Male employees show slightly greater loyalty compared to female employees”, 2; “Male and female employees show no significant difference”, 3; “Female employees show slightly greater loyalty compared to male employees”, 4; “Female employees demonstrate significantly higher levels of loyalty than male employees”, 5. For RC, the following single-item scale was used: “The human resource management (HRM) system in our organization is perceived as rational and transparent.” The item was measured on a five-point Likert scale ranging from 1 = strongly disagree to 5 = strongly agree. Previous studies have demonstrated that in organizational settings, carefully designed single-item measures can still provide acceptable levels of validity and reliability for global constructs ([20]; [2]). Furthermore, as RC is an overarching cultural perception rather than a latent multidimensional construct in our model, we believe that the single-item measure offers a reasonable proxy. The gender of HR managers was measured as a binary item (0 = female and 1 = male) using the following question: “What is the gender of your primary HR manager?”

## 3. Results

This table reports the results of the hierarchical regression analyses of the main effects of HR managers’ gender on three key organizational outcomes of female employees—organizational loyalty, job involvement, and interpersonal affinity. The models controlled for organization type and firm size.

Table 1 presents the results for H1–H3, which test the main effects of HR managers’ gender on organizational loyalty, job involvement, and interpersonal affinity. The findings indicate that female HR managers have a significant positive effect on all three outcomes, supporting H1, H2, and H3.

The results reveal that HR managers’ gender was a consistent and statistically significant predictor across all three outcomes, with beta coefficients of 0.176 for both loyalty and affinity and 0.278 for job involvement (*p* < 0.001 for all). These findings indicate that female HR managers have a strong positive effect on the organizational experiences of female employees. This supports the notion that gender congruence in managerial roles enhances employee trust, motivation, and emotional bonding, aligning with the social identity and similarity-attraction theories.

Organization types also emerged as a significant predictor of organizational loyalty (*β* = 0.107, *p* < 0.05) and job involvement (*β* = 0.171, *p* < 0.001), implying that certain types of firms—perhaps those with more inclusive or developmental cultures—foster better employee engagement. However, firm size was not a significant predictor in any of the models, indicating that firm scale alone may not meaningfully impact female employees’ perceptions in these areas.

Model fit statistics reveal moderate levels of explained variance, especially for job involvement (R^2^ change = 0.625 ***), implying that managerial gender is a strong predictor of engagement at work.

Table 2 presents the hierarchical regression results of the interaction effect between HR managers’ gender and RC on organizational loyalty. This analysis investigates whether RC moderates the relationship between a manager’s gender and organizational loyalty. Table 2 reports the results for H4, which examines whether rationality culture (RC) moderates the relationship between HR managers’ gender and organizational loyalty. The moderation effect was marginally significant (*p* = 0.066), indicating partial support for H4. The moderation effects for job involvement and interpersonal affinity (H5 and H6) were not significant.

Model 1 includes the organization type, firm size, and HR managers’ gender. As in Table 1, HR managers’ gender (*β* = 0.278, *p* < 0.001) and organization type (*β* = 0.171, *p* < 0.05) were significant, whereas firm size was not. This baseline model indicates that HR managers’ gender and organizational characteristics exert positive effects on organizational loyalty. In Model 2, RC was added and found to be a significant positive predictor (*β* = 0.196, *p* < 0.01), implying that a structured, logic-driven organizational climate contributes to higher job involvement. The gender of HR managers remained significant (*β* = 0.326, *p* < 0.001), indicating its stable effect regardless of RC. Model 3 introduced an interaction term (RC × HRMG) to test the moderation hypothesis.

The interaction effect was marginally significant (*β* = −0.442, *t* = −1.845, *p* = 0.066), implying that the positive effect of HR managers’ gender on organizational loyalty may weaken in highly rationalistic organizational environments. This nuanced finding implies that in rigid, rule-oriented settings, the benefits of gender congruence between a manager and employee can be dampened, perhaps due to limitations on relational or emotional expression. Despite the interaction not reaching full significance, the observed trend provides partial support for H4 and highlights the conditional nature of HR managers’ gender effects depending on the organizational context.

The interaction effect between RC and HR managers’ gender on the dependent variable, job involvement, was not statistically significant (*B* = −0.089, *p* = 0.150). This indicates that RC does not significantly moderate the relationship between HR managers’ gender and job involvement. Therefore, H5 is not supported.

The moderation effect was also not observed when interpersonal affinity was used as the dependent variable (*B* = −0.062, *p* = 0.445). This indicates that RC does not significantly moderate the relationship between HR managers’ gender and interpersonal affinity. Therefore, H6 is not supported.

## 4. Discussion

This study examined how HR managers’ gender and RC interact to shape key employee perceptions, focusing on organizational loyalty, job involvement, and interpersonal affinity. While the main effects (H1–H3) were statistically significant, it is important to note that they accounted for only a modest proportion of variance (low R^2^ values). This indicates that other unobserved factors may also contribute to female employees’ organizational outcomes and should be examined in future research. Nevertheless, the findings consistently highlight the positive impact of female HR managers, particularly in fostering emotional connection, psychological engagement, and organizational commitment among female employees. H1–H3 were supported, demonstrating that female HR managers significantly enhance loyalty, job involvement, and interpersonal affinity. These results align with social identity theory, which suggests that shared gender identity promotes trust, psychological safety, and stronger workplace engagement ([5]; [19]).

However, H4–H6, which proposed that RC moderates these relationships, received only partial support. Specifically, the moderation effect of RC on organizational loyalty (H4) yielded a *p*-value of 0.066, which does not meet conventional thresholds for statistical significance. Rather than treating this as confirmatory evidence, we interpret it as a marginal trend that warrants further investigation. This suggests that in bureaucratic, logic-driven cultures, the positive influence of female HR managers on loyalty may be weakened. This observation aligns with feminist organizational critiques ([1]; [13]), which argue that gender power dynamics are reinforced by institutional norms privileging rationality and formalism over emotional and social competencies. In such environments, contributions related to inclusiveness, fairness, and interpersonal sensitivity may be undervalued, thereby reducing their impact on formal outcomes such as organizational loyalty.

Another limitation of this study concerns the measurement of rationality culture (RC), which was assessed using a single-item scale. While this approach was necessitated by the constraints of the dataset, we acknowledge that multi-item scales are generally preferable for capturing complex and multidimensional constructs. Future research should employ validated multi-item measures of RC to improve construct validity and to better capture its nuanced dimensions within organizational contexts. This would allow for a more robust examination of how RC interacts with other variables such as gender representation and employee outcomes.

Interestingly, RC did not significantly moderate the relationships between HR managers’ gender and job involvement or interpersonal affinity. A possible explanation is that day-to-day HR interactions exert influence independent of the broader organizational culture. Employees’ perceptions of interpersonal affinity and engagement may be shaped by direct interactions and psychological signals, which can operate even within highly rationalistic or formalistic environments.

These results can also be better understood through the theoretical frameworks underpinning this study. Specifically, the consistently positive effect of women HR managers on organizational loyalty, job involvement, and interpersonal affinity aligns with social identity theory, which suggests that shared gender identity fosters trust, belonging, and psychological engagement in the workplace. Even in highly formalized organizations, the salience of identity congruence appears to support relational dynamics and attitudinal commitment.

Meanwhile, the marginally significant moderation effect of RC on organizational loyalty—and its absence in the other outcomes—can be interpreted through feminist organizational theory. This perspective highlights how institutional cultures rooted in rationality and bureaucratic logic tend to marginalize emotional labor and relational leadership, limiting the influence of women HR managers in strategic domains. Although women may occupy leadership roles, structural norms may constrain their ability to convert interpersonal trust into systemic impact. This supports feminist critiques that emphasize symbolic inclusion without real power redistribution.

By explicitly mapping these empirical patterns onto both theoretical lenses, the findings reinforce a dual-level understanding of gendered HR leadership—where micro-level identity alignment fosters trust, but macro-level cultural constraints may inhibit structural influence. This deeper interpretive framing enhances the contribution of this study to gender, HRM, and the organizational theory literature.

Another explanation relates to the distinct nature of the outcome variables. Although organizational loyalty reflects a formal, long-term alignment with institutional goals and is thus more sensitive to the cultural context, interpersonal affinity and job involvement are often rooted in localized workplace experiences and peer-to-peer dynamics. Therefore, these latter outcomes are less likely to be disrupted by overarching structural formalism.

It is also important to consider the measurement limitations. In this study, RC was assessed using a single item, which may not have fully captured the nuanced variations in the cultural context. Without robust, multidimensional measures, detecting subtle interaction effects becomes statistically challenging, particularly for psychological constructs such as job involvement and interpersonal affinity.

Finally, the sample characteristics and contextual factors may have influenced the findings. The Korean Women Managers Panel Data primarily cover large organizations (with 100 or more employees), where departmental subcultures may dilute the broader organizational culture’s effects. Additionally, industry differences, such as the contrasting environments of manufacturing, services, or public administration, may further obscure the moderating role of RC, making its impact more context-specific.

Overall, the findings highlight that HR managers’ gender is a robust and positive predictor of female employee outcomes, regardless of broader cultural constraints. However, the moderating role of RC appears limited or conditional, primarily affecting formal constructs such as loyalty while exerting minimal influence on relational or motivational outcomes. This study contributes to the growing literature demonstrating that gender-congruent HR representation can enhance employee trust, engagement, and organizational commitment, but the full realization of these benefits may depend on organizational environments that recognize and value inclusive and supportive HR practices.

### 4.1. Implications

Theoretically, this study offers preliminary insights into how gender representation in HR leadership interacts with organizational culture to influence women employees’ experiences. While the findings robustly support the main effects of HR managers’ gender on organizational loyalty, job involvement, and interpersonal affinity, the evidence for the moderating role of rationality culture (RC) is more tentative. Specifically, only one interaction effect—on organizational loyalty—was marginally significant, and no significant moderation was observed for job involvement or interpersonal affinity.

These mixed results suggest that the conditional nature of gender representation in HR may be more context-sensitive and nuanced than originally theorized. Rather than confirming a strong and uniform moderating effect of RC, the findings indicate that some elements of employee attitudes—particularly those linked to formal organizational commitment—may be more vulnerable to structural constraints than others rooted in interpersonal dynamics.

Thus, while the study integrates feminist organizational theory and social identity theory to frame its hypotheses, the results suggest only partial support for their interaction, reinforcing the need for more empirical research that can unpack the layered effects of gender, power, and culture in HR contexts. Future studies may benefit from exploring alternative cultural dimensions, such as organizational support, leadership climate, or informal norms, that could mediate or moderate the relationship between gender representation and workplace outcomes more strongly than RC alone.

Beyond theoretical contributions, the study offers practical insights for HRM policy and strategy. Recognizing that female HR managers are not a monolithic group, we caution against overly generalized assumptions about their leadership style or influence. Variations in authority, organizational support, and leadership orientation significantly shape their ability to implement inclusive practices. As such, organizations seeking to advance gender equity in HRM must also consider the institutional logics and power structures that may constrain female leaders’ effectiveness.

Moreover, relational competencies—often associated with female leadership—should not be viewed as secondary or soft skills but rather as critical elements of organizational success. HR development frameworks should incorporate and validate these competencies formally, regardless of the gender of the leader. This approach not only promotes inclusivity but also ensures that emotional intelligence, mentorship, and trust-building become integral to performance-oriented HR systems.

### 4.2. Recommendations and Future Research

The findings of this study offer several practical recommendations for organizational leaders and HR practitioners. First, organizations should critically assess their internal cultural frameworks, particularly those that overemphasize rationality, formalism, and control at the expense of relational and supportive HR practices. Such cultural biases may unintentionally suppress the positive effects of female HR representation by privileging procedural logic over human-centered practices. To fully leverage the potential of gender-congruent HR presence, organizations must strive to create balanced environments that value both efficiency and interpersonal connection.

Second, HR development initiatives should move beyond technical and administrative training to include programs focused on building inclusive, fair, and supportive practices. Emotional intelligence, ethical awareness, individualized support, and psychological safety are essential components of effective HR management, particularly for female HR leaders navigating traditionally masculine or bureaucratic organizational contexts.

Third, organizations are encouraged to conduct regular audits or internal assessments of their institutional norms and evaluation criteria. These assessments should investigate whether existing performance metrics, promotion pathways, or cultural expectations unintentionally reinforce gender power dynamics that devalue interpersonal contributions. Addressing these structural barriers is critical to ensuring that female HR managers are empowered to exercise meaningful authority and positively influence employee outcomes.

Future research can expand on the insights of this study through several promising avenues. Longitudinal designs would enable scholars to explore how perceptions of female HR leadership evolve, particularly in relation to organizational changes, leadership transitions, or shifting cultural norms. Qualitative approaches, such as in-depth interviews or focus groups, can offer richer insights into how female employees interpret the symbolic and practical dimensions of HR representation in rationalistic work environments. Additionally, cross-cultural comparative studies would be valuable in testing whether the conditional effects of RC can be generalized across different national, sectoral, or institutional contexts, enhancing the external validity of the findings. Finally, incorporating intersectional variables, such as age, ethnicity, tenure, and parental status, can illuminate how multiple social identities interact to shape employee perceptions of HR, offering a more comprehensive and inclusive perspective.

## 5. Conclusions

This study examined how HR managers’ gender and RC jointly influence female employees’ perceptions of organizational loyalty, job involvement, and interpersonal affinity. Drawing on feminist organizational theory and social identity theory, the findings demonstrate that female HR managers positively shape female employees’ emotional connection, psychological engagement, and organizational commitment. The results confirm that gender representation in HR can foster enhanced trust, fairness perceptions, and workplace satisfaction among female employees.

The study also revealed that the organizational cultural context partially moderates these effects. Although RC marginally weakened the positive relationship between female HR managers and organizational loyalty, it exerted a limited influence on job involvement and interpersonal affinity. These findings imply that although bureaucratic norms can dampen the impact of gender representation on formal organizational commitments, they do not necessarily constrain day-to-day relational and motivational dynamics. Thus, this study advances a more nuanced understanding of how HR managers’ gender and organizational culture jointly shape female employees’ attitudes and highlights the context-dependent nature of inclusive HR practices in formalized workplaces.

## Figures and Tables

**Table 1 behavsci-15-01088-t001:** Hierarchical regression results: main effects of HR managers’ gender on organizational outcomes.

Variable	Dependent Variable					
	Organizational Loyalty		Job Involvement		Interpersonal Affinity	
	Β	*t*-Value	*β*	*t*-Value	Β	*t*-Value
Organization type	0.107 *	2.001	0.171 ***	3.294	−0.002	−0.030
Firm size	0.052	0.983	0.025	0.478	0.016	0.307
HR gender	0.176 ***	3.297	0.278 ***	5.369	0.176 ***	3.285
R^2^	0.041		0.096		0.031	
Adjusted R^2^	0.032		0.088		0.023	
∆ R^2^	0.485		0.625		0.636	
F value	4.846 *		12.113 ***		3.659 *	

Notes. *β* = standardized coefficient; n = 346. Organization type: private sector = 0; public sector = 1. Firm size: <100 employees; 1 = 100~299; 2 = 300~999; 3 = 1000~1999; 4 = ≥2000 employees. HR gender: 0 = female, 1 = male. * *p* < 0.05; *** *p* < 0.001.

**Table 2 behavsci-15-01088-t002:** Moderating effect of RC on the relationship between HR managers’ gender and organizational loyalty.

Variable	Model1		Model2		Model3	
	*β*	*t*-Value	Β	*t*-Value	*β*	*t*-Value
Organization type	0.171 *	3.294	0.145	2.811	0.144	2.805
Firm size	0.025	0.478	−0.005	−0.103	0.001	0.021
HR gender	0.278 ***	5.396	0.326 ***	6.197	0.738 ***	3.217
Rationality culture			0.196 ***	3.649	0.474 **	2.964
RC × HRMG					−0.442 †	−1.845 †
R^2^	0.031		0.036		0.038	
∆ R^2^	0.023		0.025		0.024	
F value	3.659 *		3.209 *		2.681 *	

Notes. *β* = standardized coefficient. n = 346. Organization type: private sector = 0, public sector = 1. Firm size: <100 employees; 1 = 100~299; 2 = 300~999; 3 = 1000~1999; 4 = ≥2000 employees. HR gender: 0 = female, 1 = male. † *p* < 0.10; * *p* < 0.05; ** *p* < 0.01; *** *p* < 0.001.

## Data Availability

Data are contained within the article.

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
