# Peer review of "HR Managers’ Gender and Rationality Culture: Interaction Effects on Female Employees’ Workplace Outcomes"

_behavsci, 2025, doi:10.3390/bs15081088_

Round 1

Reviewer 1 Report

Comments and Suggestions for Authors

Thank you for inviting me to review this paper. The article is intersting and relevant. However, I have some concerns which may improve the quality of the paper.

First, the research motivation and signficant need improvement. I would suggest adding practical examples to explain the value of this study. Moreover, while this study identifies a significant gap in understanding how the gender of HR managers interacts with rationality culture (RC) within organizations to shape female employees’ outcomes. Authors should more epxlicitly explore what is new and important that we don't know, thereby explaining critical research gap.

Second, while the study integrates feminist organizational theory and social identity theory to explain the mechanisms by which female HR managers can foster trust, involvement, and loyalty among female employees. However, these theories are superficially explained. I suggest provide logical jusification of why two theories are necessary to develop the model and how these theories integrate to explain theory, hypotheses and contributions of the study. 

Third, while this research used a single-item measure for RC, it raises concerns about construct validity, especially given its central role as a moderator. This is a critical limiation of the methods. A multidimensional or validated scale would have been more appropriate to capture the nuances of rationality culture. Authors are suggested to provide the justification of the measure. In addition, study should provide validatity and reliablity statistics of the measurement model.

I hope my comments are helpful in improving the paper.

Reviewer 2 Report

Comments and Suggestions for Authors

Thank you for the opportunity to review this manuscript. The topic addressed is relevant and timely, especially considering the growing interest in gender dynamics, organizational culture, and HRM practices. The study is conceptually grounded in feminist organizational theory and social identity theory, and it aims to provide valuable insights into how the gender of HR managers interacts with rationality culture to affect female employees’ workplace outcomes. However, there are several areas that require clarification or improvement before the paper can make a strong scholarly contribution.

1. Theoretical framework and coherence

  • The manuscript presents multiple theoretical lenses (feminist organizational theory, social identity theory, role congruity theory), but their integration is not entirely clear. For instance, while the abstract emphasizes feminist organizational theory, the introduction prioritizes social identity and role congruity theories. I recommend better alignment and clarification across sections.

  • The discussion of rationality culture (RC) is conceptually rich but lacks empirical anchoring. No prior studies applying RC to HR managers are cited, and the application of feminist critiques to HR-specific dynamics is mostly theoretical.

2. Measurement issues

  • One of the main weaknesses of the manuscript lies in its operationalization of key variables:

    • RC is measured using a single-item, unvalidated scale, which significantly undermines the construct validity of the moderating variable.

    • Outcome variables (loyalty, involvement, affinity) are assessed via comparative items between male and female employees, which does not directly capture female employees’ own perceptions or experiences.

    • The origin of these measures is not cited. It is unclear whether they are adapted from validated instruments or created ad hoc. This lack of methodological transparency is problematic.

3. Sample and generalizability

  • The use of data from the Korean Women Managers Panel Survey is a strength in terms of scale. However, the manuscript does not justify the use of the 2018 wave nor does it clearly describe the respondents' profile (e.g., are they HR professionals, general employees, or organizational representatives?).

  • The representativeness and contextual differences across sectors are acknowledged in the discussion but not addressed methodologically.

4. Statistical analysis and interpretation

  • The main effects (H1–H3) are statistically significant but explain only a modest amount of variance (low R² values), which should be acknowledged in the discussion.

  • The only partially supported moderation effect (H4) has a p-value of .066, which does not meet standard thresholds for statistical significance. The discussion overstates this result, presenting it as “support” rather than “marginal trend.”

  • The authors should be more cautious in their interpretation and avoid generalizations not fully sustained by the data.

5. Implications and contribution

  • The theoretical implications are presented as if the findings strongly support the conditional nature of gender representation in HR, when in fact only one moderation effect was marginally significant.

  • The practical implications are reasonable but assume a generalized profile of “female HR managers” that may not reflect diversity in leadership styles, levels of authority, or organizational settings.

6. Suggestions for improvement

  • Clearly state the source or justification of all scales used.

  • Consider using validated, multidimensional instruments for RC and employee outcomes in future iterations.

  • Distinguish more carefully between perceived symbolic representation and actual influence of HR managers in different organizational contexts.

  • Moderate the tone of theoretical and practical implications in accordance with the strength of the empirical findings.

In summary, the manuscript addresses an important topic with theoretical potential. However, improvements in construct operationalization, statistical interpretation, and theoretical coherence are necessary to strengthen its academic contribution.

Reviewer 3 Report

Comments and Suggestions for Authors

The paper is well written in terms of content and form. The introduction provides a clear description of the context of the paper, before justifying the study, i.e.  to address the gap in the body of knowledge by examining how an HR manager’s gender and RC jointly influence female employees’ perceptions and attitudes. The study explores whether the gender of HR managers enhances interpersonal affinity, job involvement, and organizational loyalty and whether the cultural context of rationality moderates these effects.

The methodology is appropriate for the purpose of the study. The research design, questions, hypotheses and methods are clearly stated. The sample selection rationale is clear, and the procedure used to collect data is very clear and logical. The results are presented and interpreted well.

The study has contributed to scholarship by demonstrating that although bureaucratic norms can dampen the impact of gender representation on formal organizational commitments, they do not necessarily constrain day-to-day relational and motivational dynamics.

This paper is ready for publication.

Comments on the Quality of English Language

The use of English is very good

Author Response

We sincerely thank the reviewer for the positive and encouraging feedback on our manuscript. We greatly appreciate your recognition of the study’s contributions and your recommendation for publication.

Reviewer 4 Report

Comments and Suggestions for Authors

The article "HR Manager’s Gender and Rationality Culture: Interaction Effects on Female Employees’ Workplace Outcomes" presents an interesting investigation into how the HR managers’ gender and organisational rationality culture shapes employees' experiences at work. The topic is both important and highly relevant to the journal’s focus, offering valuable insights into organisational culture, and their impact on workplace outcomes. cs. I have provided a number of comments and suggestions below, which I hope will be helpful in strengthening the manuscript further.

Introduction:

  • On page 1, authors claims that the topic is underdeveloped, but no evidence is presented. Is there research on the similar topic on the other areas such as leadership, other managerial levels, such as supervision in organisations, etc.?
  • It would be beneficial to also explain the paper contribution and significance in the introduction after explaining the gap- why it is important along with potential significance and implications.

Literature:

  • Page 3, last paragraph please provide appropriate reference/citation, particularly for the last line of the paragraph (page 3) where explain “This perceived misalignment can create barriers 91 to perceived legitimacy and authority for women…” also unpack the sort of barriers it may create.
  • The RC section under literature review is under-cited. Please provide proper, relevant and recent citations to support the statements and claims made in this section.
  • The first paragraph under feminist theory: please provide evidence for “Gendered assumptions about who is a “natural” authority figure persist in many workplaces, resulting in a bias toward male-dominated leadership even when women are present in significant numbers.” What are some examples of these organisations? Please provide research studies done in the space that shows the persistence of these “norms” particularly where women outweigh men.
  • I’d suggest using more recent studies indicating “the presence of women in managerial roles may be used by organizations to signal gender inclusivity…” the study was done 25 years ago.
  • social identity theory, mentioned in the abstract, but is not properly explained under section 1.1. Literature review. It was very briefly mentioned on page 4 in the second paragraph.
  • I’d recommend avoiding repetition. For instance, “a rationality culture, characterized by formalism, logic, and depersonalized decision-making (Weber, 1947)”repeated in literature review (pages 3 and 4)

Methods:

  • I’d recommend authors include an introductory paragraph explaining their methodological approach and its appropriateness for this research study.
  • Under sample section, please specify the years during which the data was collected, biennially from 2007 until what year? Was it until 2018 as mentioned in line 202, page 5?
  • Were these data collected from women managers only? If not, then please clarify. If yes, please explain why the data collected from women managers only and how this may impact the result of the survey.
  • Please specify the type and/or sector of these 769 companies, e.g. education, military, corporate, etc.?
  • Please also provide more information about participants, such as their age range, year of employment as the manager and other information if applicable.
  • While I appreciate the detailed explanation of the measurements, the data analysis is missing under method section. There is also a need for explaining how the data was analysed and the rationale behind it, including any tools used.

Results:

  • In relation to table 1 and 2, authors explain the effect of organisation’s type and characteristics on loyalty, but it’s not clear what these types and characteristics are- this part is vague and needs to be clarified in the methods section or explained in the results section.
  • In the results section, please specify what hypothesis these data are presenting, for instance table 1 what hypothesis is being tested and how what the results are showing to affirm or reject each hypothesis?

Discussion:

I appreciate the authors’ inclusion of the study’s limitations, implications, and recommendations, which adds clarity and transparency to the work. However, the discussion section would benefit from further revision. Currently, it lacks a deeper level of critical analysis and does not sufficiently engage with the theoretical framework outlined earlier in the paper, particularly feminist organisational theory and social identity theory. Strengthening the connection between the findings and these theoretical lenses would enhance the interpretive depth of the discussion and provide a more robust contribution to the literature.

General recommendation:

  • Using more inclusive language, such as women instead of female- referring to the gender and social identity rather than physiological sex
  • It would be beneficial to indicate the context in which the research was conducted, particularly the geographical or organisational setting, in the abstract. This would help readers better understand the relevance and applicability of the findings.

Round 2

Reviewer 1 Report

Comments and Suggestions for Authors

Thank you for allowing me to read this paper. While I commend authors effeort in addressing my concerns. I am satisfied with the response. I have a minor suggestion. Improve the citations by including recent references of 2025.

Author Response

We appreciate your helpful feedback. In response, we have added three recent references from 2024–2025 to the Literature Review and References section to enhance the currency of the manuscript. These additions are highlighted in sky blue for your review.

Reviewer 4 Report

Comments and Suggestions for Authors

Thank you for your thoughtful and thorough revisions. I can see that you have taken on board the majority of the comments and suggestions provided in the earlier round, and the manuscript has been significantly improved as a result. The revised version shows greater clarity, coherence, and depth. I believe the paper is now suitable for publication. One suggestion for further strengthening the manuscript is to more clearly and consistently link the findings and discussion to the theoretical framework underpinning the study, which would enhance the overall contribution and alignment of your work.

Author Response

Thank you for your insightful suggestion. We have revised the Discussion section to better emphasize the alignment between our findings and the theoretical framework, particularly highlighting how feminist organizational theory explains the marginal moderation effects. The added content is marked in sky blue in the revised manuscript.